# Not only a territorial matter: The electoral surge of VOX and the anti-libertarian reaction

**Rodrigo Ramis-Moyano** [1,2] *, **Sara Pasadas-del-Amo**[1], **Joan Font**[1]

**1** Instituto de Estudios Sociales Avanzados, Consejo Superior de Investigaciones Científicas (IESA-CSIC), Córdoba, Spain, **2** Área de Sociología, Departamento de Ciencias Sociales, Filosofía, Geografía y Traducción e Interpretación de la Universidad de Córdoba (UCO), Córdoba, Spain

* rramis@iesa.csic.es

**Data Availability Statement:** All PACIS files are available from the Open Access institutional repository of the Spanish National Research Council (https://doi.org/10.20350/digitalCSIC/14804).

## Abstract

Although previous work has shown the complexity of motives behind the VOX vote, its emergence is often associated mainly with the Catalan conflict. Our analysis shows that VOX's first electoral success was marked importantly by preferences related to territorial conflict, but also by opposition to immigration, authoritarianism or ideology. The main contribution of the paper lies in demonstrating something that until now had not been empirically verified: the relevance of anti-feminist attitudes amongst the VOX electorate. This shows how, since its onset, these voters have not been so different from voters of other European radical right-wing parties, and how VOX has channelled into elections the reaction against different expressions of a more diverse and egalitarian society.

## Introduction

Until 2018, Spain was one of the few countries in the European context in which there was no relevant political party on the radical right [1]. Today, although this singularity has disappeared and VOX has established itself with a strong parliamentary presence at all territorial levels, certain exceptionalism appears to remain when we try to understand and place this party and its electorate in comparative perspective. Much of the international analysis of the radical right emphasises a broad range of explanations, including the "losers of globalisation" and economic insecurity thesis [2], grievances and negative feelings towards immigration [3] or the rejection of cultural change [4]. However, in the Spanish case, most of the existing literature on VOX seems to point towards a considerable singularity. According to this, VOX's electoral surge was in its beginnings mainly the product of a reaction of Spanish nationalism towards the independence threat that emerged in Catalonia [5, 6]. Even the book of Rama *et al* [7]–the most comprehensive account of VOX to date–continue to attribute VOX's electoral emergence mainly to the Catalan conflict [7: 93].

Our argument focuses on the fact that this explanation is incomplete. Many empirical analyses on VOX's electoral surge are based on survey data from "Centro de Investigaciones Sociológicas" (CIS), which is the main source of information on voting behaviour in Spain. However, its questionnaires lack important variables that would allow testing some of the alternative explanations for VOX's electoral support. This led to the idea that the Catalan

**Funding:** The survey was conducted using the Citizen Panel for Social Research (PACIS), a probability-based mixed-modes panel of the Andalusian population self-financed by the Instituto de Estudios Sociales Avanzados (IESA-CSIC) with funding of the Consejo Superior de Investigaciones Científicas (CSIC) under the program "Proyectos Intramurales" (ref. 201710E018). Rodrigo Ramis Moyano is beneficiary of the University Teacher Training Program (FPU2019) funded by the Spanish Ministerio de Ciencia, Innovación y Universidades. The funders had no role in study design, data collection and analysis, decision to publish, or preparation of the manuscript.

**Competing interests:** The authors have declared that no competing interests exist.

conflict was almost the only relevant explanatory factor for its rise. Our analysis shows that other common explanations of the Western European Radical Right (authoritarianism, the rejection of immigration and the reaction against cultural liberalism) are also an essential part of VOX's electoral success since its onset. Its interpretation is therefore much more complex– and, at the same time, less unique in a comparative perspective–than it might seem.

To do this, we use an original dataset, specifically designed to test if the main explanatory factors for the rise of other Western European radical right-wing parties apply to the election where VOX got represented in a parliament for the first time in Spain: the 2018 Andalusian regional election. The results–using two alternative measures of support for VOX–show that the preferences and attitudes related to the territorial conflict do matter, but their relevance is clearly reduced when we add alternative explanations. Our main contribution to this strand of research is twofold. First, our analysis provides empirical evidence on the importance of some factors for which there were limited data in Spain: authoritarian attitudes and opposition to feminism, with the latter showing an important explanatory power of support for VOX, even after controlling for other alternative factors. Second, by being able to model the different factors in the same dataset, we can rank them according to their relevance explaining the vote for VOX in that particular election. As mentioned, this is something that can't be done using CIS post-election surveys because they lack variables that contribute to explain the vote for individual parties, particularly the most recent ones such as VOX.

The following section examines what we know on the subject in comparative perspective, helping us to develop our hypotheses. This is followed by a brief description of the context surrounding the election analysed, and then an explanation of the data and methods used in the article. The final two sections present the analysis and results, as well as a final discussion of how the interpretation of the Spanish case fits into the analysis on the Western European Radical Right.

## The vote for the Western European Radical Right: The Spanish case in comparative perspective

In the weeks after the 2018 Andalusian Parliament elections, several analyses published in the press pointed out to the role of different factors explaining the vote for VOX such as anti-immigration attitudes [8–10] or the penalty of corruption in the regional government and the rejection of the gender equality and historical memory acts [10]. Also, more recent and complete contributions have demonstrated that opposition to immigration [7, 11–14], cultural conservatism or attitudes towards gay marriage, as well as the preference for an authoritarian regime [7] or populist attitudes [13] are factors that help driving electoral support for VOX, suggesting some similarities with other Western European radical right-wing parties.

Nonetheless, research on VOX seems to point to a remarkable exceptionality when we come to its rise, linked to a factor which is specific to the Spanish case: the territorial crisis. According to this thesis, the vote for the Spanish far-right went from one singularity (being almost non-existent) to another (being mostly explained in territorial terms, compared to the plurality of explanatory factors in other countries). In contrast, we expect that most of the reasons that have prompted the appearance and success of radical right-wing parties in neighbouring countries should also be present in the Spanish case since its inception. Thus, focusing on the empirical analysis of the Spanish case, we contribute to the debate on the *demand-side* of *European Radical Right Studies* [15].

Following the views of Ferreira [16], we define VOX as a radical right party, which is a subcategory of parties of the far-right: those at the right of the classic Christian-Democratic and Conservative families. The support for the radical right in Western Europe appears to be

linked to a large number of explanatory factors that can be summarized in six major blocks: the cultural reaction against the "libertarian left", the nationalist explanation, the anti-immigration reaction, the authoritarian explanation, the populist protest and the economic explanation. Over the following pages we present the main reasons why each of them may be relevant, as well as a plausible explanation of how they fit into VOX's electoral surge. We claim that the first five have signs of plausibility for the Spanish case, and therefore are the ones that will be empirically tested in this study. While testing the role of all of them our main contribution lies in the consideration of VOX as an anti-libertarian reaction, focusing on the issue of feminism, the strongest (and most contested by radical right) social movement of the libertarian side in the recent Spanish reality [17, 18].

Far-right's defence of traditional values represents a rejection of (and a reaction against) "post-materialist" moral and cultural changes, such as LGBT+ rights, abortion, or the role of women in social and political life, among others [4]. Whereas in some Western European countries, such as Belgium or the Netherlands, this debate does not seem to be central to understanding the success of the radical right [19], in others such as Poland, it may be an important explanatory factor [20]. In the Spanish case, there is also conflicting evidence available. Rama *et al* [7] have shown the relevance of attitudes towards Gay marriage whereas the combined set of items used by Marcos-Marne *et al* [13] does not show explanatory power when controlling for alternative explanations. In any case, the direct test of any variable explicitly measuring opposition to feminism is quite limited for the Spanish case.

This factor is worthy of special focus in our analysis considering that the Andalusian regional election, the first electoral success of VOX, was held at the end of 2018, the so-called year of the women. On March 8th that year, massive rallies were held in most capital cities of Spain, and more than 5 million Spaniards–according to main Spanish unions–participated in the first nationwide feminist strike. These events were fuelled by the #MeToo movement, but also as a consequence of the public outrage generated by different cases of sexual violence against women–and its subsequent judicial treatment–that had taken place in Spain. The Spanish Women's Movement proved its strength with further demonstrations throughout the year [21, 22]. In this context, VOX built its campaign for the Andalusian election along their rejection of the Law against Sexist Violence and the exaltation of the traditional role of women in the family, among other subjects [18, 23].

Opposition to gender-based policies were present in VOX's foundational manifesto and party statutes since its inception [23] and explicit anti-feminist discourse was deployed in official parliamentary speeches made by representatives of VOX [24] and posted in their social media accounts on the occasion of the election campaign in Andalusia [25] and 2019 national and other regional elections [18, 23]. A content analysis of 73 parties' manifestos concurring to Spanish regional elections in 2019 showed that, in fact, VOX stays alone when it comes to include women-related regressive policy proposals in their political manifestos [26]. Therefore, it seems plausible to consider this factor as a further explanatory element of the reasons for the electoral emergence of VOX.

*Hypothesis 1*: Opposition to the values of the libertarian left, particularly that which has been most successful in recent years in the case of Spain, feminism, is positively related to a predisposition to vote for VOX, even after controlling for other explanatory factors.

However, other alternative hypotheses are also likely to be present in the explanation of voting for VOX. Nationalism is not one of the most common comparative explanations for the vote to the radical right [27]. Nevertheless, it has one of the highest rates of empirical verification (100%) where it has been tested [e.g. 28]. "Nativism"–a combination of nationalism and xenophobia–would also be a starting point for the defence of a strong and homogeneous national identity that

opposes any threat to it, whether from outside or from within. In the Spanish case, the first empirical study on the subject placed most of the explanation for the vote for VOX (along with ideology) on the territorial factor, specifically in the defence of fewer powers for the regions [5]. Subsequent work has maintained this argument [6], although Turnbull-Dugarte and others [29] have enriched it by showing that the effect of nationalist sentiment is conditioned by the perception of the political situation, occurring only when the assessment of the latter is negative. In any case, this line of argument follows Liñeira [30] and his evidence showing how the territorial conflict has been deepening in the Spanish case and is now probably part of the explanation for the surge of new parties both at the right and left side of the political spectrum [31].

Attitudes towards immigration and ethnic-cultural diversity are probably among the most studied reasons for analysing radical right's vote [4] and include a wide variety of sceptical and negative feelings towards others [32]. These are key factors for voting for many of these parties, both in Nordic, and central and Mediterranean European countries [3, 33]. However, this relationship is far from unanimous and appears only in part of the cases [27]. Previous Spanish research shows contradictory findings, with some studies pointing to a null relationship [5, 6] whereas the most recent pieces show that immigration plays an important explanatory role [7, 13]. With immigration being an issue highly salient for VOX's voters [7, 10], and increasingly discussed [12, 14] and polarised [34], the great presence of the problematisation of immigration in the party's discourse [7, 13, 16, 35] suggest that we should indeed be able to find a relationship between these attitudes and their electoral support.

Altemeyer [36] offers one of the most precise definitions of right-wing authoritarianism, combining traditionalism, attachment to authority and hostility towards "the others", which results in support for non-liberal views of democracy [37]. We focus on its support for law, order, and the fight against crime as its most distinctive aspect. In practice, many of the parties in this political family establish a very clear link between levels of insecurity and immigration [32, 38]. At European level, little attention has been paid to this factor, but some studies have shown a positive relationship between authoritarianism and voting for the radical right [28, 33, 39]. Marcos-Marne *et al* [13] have recently argued that it plays a role for the Spanish case, a quite likely scenario since authoritarian aspects are also present in the VOX electoral discourse [7, 16, 35]. This matters because, as shown by Rama *et al* [7: 129–130], support for VOX increases among those individuals who prefer an authoritarian regime or who consider that the type of political regime does not matter.

The analysis of the populist component in this political family is shaped by the lack of clarity and homogeneity in its definitions and operationalisation [40, 41]. In this explanation, we include the idea of a protest vote and a global amendment to the functioning of the political system. According to Stockemer and others [27], in the few instances that this factor has been analysed, its explanatory capacity for voting for this group of parties is very high (71% success). Turnbull-Dugarte [5] and Mudde [42] include VOX among populist parties, while Norris [43] classifies it as the most populist of the Spanish parliamentary parties. However, other empirical analyses go against this idea. Marcos-Marne *et al* [13], using the Akkerman *et al* [44] populism scale, find mild-low levels of populism in VOX discourses, playing a much more discreet role than nationalism or authoritarianism. Ferreira [16] also indicates that the populist component in the discourse of this party, despite being present, is not explicit. When it does appear, it is always tangential and subordinate to nationalist rhetoric. The specific role of one of the central populist ideas, the opposition of people versus elites–to which many authors attribute capital importance [45]–is used here.

Finally, the extensive literature on the economic thesis does not offer a unique explanation for the far-right vote. While some authors provide evidence supporting the "losers of globalisation" thesis [2], others point out the variability of profiles of their electorate across countries

[46] or indicate that the explanation has more to do with the insecurity of a possible loss of 'status' in an adverse economic context [47]. For the Spanish case, available data shows that high-income individuals are more prone to vote for VOX than low-income [7, 29]. This relationship can most likely be explained through double reasoning. In terms of *supply-side*, VOX's economic discourse has focused much more on reducing the tax burden [16], making it more attractive among the richest sectors. Beside this, the strong link between blue-collar workers and left-wing parties [48: 381] could be less negatively affected than in other Western European countries, making them more difficult to capture through new political options. Since in those political contexts where the economic issue is relevant and polarised–as in Spain [34]–there is a negative correlation of the working-class vote for the far-right [49], we do not expect to find a positive relationship between economic deprivation and the vote for VOX. Ortiz [6], Turnbull-Dugarte *et al* [29] and Rama *et al* [7: 108] have shown that the hypothesis of the "losers of globalisation" does not work in explaining the vote for this party.

Therefore, we incorporate the following alternative hypotheses:

*Hypothesis 2.1*: A particular concept of Spanish nationalism, expressed in the defence of a centralised state and support for hard-line policies with peripheral nationalisms, is positively related to a predisposition to vote for VOX.

*Hypothesis 2.2*: Considering immigration as a problem and having a negative attitude towards ethnic-cultural diversity is positively related to a predisposition to vote for VOX.

*Hypothesis 2.3*: Support for authoritarian values and seeing security policies as a priority are positively related to a predisposition to vote for VOX.

*Hypothesis 2.4*: Political discontent, expressed through populist proposals that place the voice of the people as a central idea, is positively related to a predisposition to vote for VOX.

## The electoral context

On 2nd December 2018, elections for the Parliament of Andalusia–the most populated region in Spain–were held. The elections were hastily brought forward after the breakdown of the agreement between Ciudadanos (liberal) and the party in the Andalusian Government at the time: PSOE. The election results included a sharp decline in turnout and the growth of new political parties on both sides of the ideological spectrum, to the detriment of the results obtained by the traditional parties of the centre-left (PSOE) and the right (PP). The elections resulted in the first change of party in government in the history of Andalusia as an autonomous region, with a new coalition government between the PP and Ciudadanos, supported externally by VOX.

These elections were also the first in which a radical right-wing party gained parliamentary representation in Spain since the 1979 general election. VOX, which in the previous Andalusian elections had obtained less than 0.5% of the votes, obtained 11% on this occasion. This important electoral growth of the party in Andalusia would be validated in the following general elections in April and November 2019, in which it obtained 10.3% and 15% of the votes respectively and has not changed substantially since then. Thus, if our results come from this specific electoral context, the level of support to VOX has not changed dramatically from there on, pointing to certain stability in its voting patterns [7].

## Data and methodology

Our analysis is based on data provided by the "Survey on the electoral behaviour of Andalusians in the regional elections of 2nd December 2018" (https://doi.org/10.20350/digitalCSIC/

14804), carried out by the Instituto de Estudios Sociales Avanzados (IESA-CSIC) in March 2019 (see Data Sheet in Table A1 in S1 File). The survey was conducted within the framework of PACIS, a probability-based panel of the general population residing in Andalusia, based on mixed modes of survey administration. This panel, inspired by the Dutch LISS panel, was recruited offline in 2015. The characteristics of the methodological design and the results of the recruitment process can be found in Arcos *et al* [50: 8–10]. Respondents who agreed to participate in the panel signed written informed consent forms. Consent was asked again with each invitation to take part in a PACIS survey. Participants in the surveys were informed that completion of the questionnaire implied their consent to participate. All data was handled applying the data protection principles of the Spanish and European Union's regulation. Ethics approvals for survey research among adults were not required by the Spanish regulation when the panel was recruited and the survey used in our analysis was conducted. In 2020, the Ethics Committee at the Spanish National Research Council required the evaluation of all projects involving human participants. The protocols of the PACIS project were reviewed and approved by the Committee later that year and have served for the surveys conducted after that date.

Regarding our survey, 1,037 of the 2,293 panellists over the age of 18 with the right to vote in the regional elections who were invited to participate in it answered to the questionnaire. This represents a response rate of 45.2%. The results of the survey have been subjected to a weighting process using four variables (sex, age, educational level, and size of the municipality) so that they faithfully correspond to the distribution of the Andalusian population they represent (see Tables A2 to A4 in S1 File).

To test the hypotheses regarding the main explanatory factors for voting for VOX, we have used a double analysis, following the approach used by Turnbull-Dugarte *et al* [29] and by Marcos-Marne *et al* [13]. First, using Linear Regression we study the effect of a series of independent variables on the declared probability of voting for VOX in a hypothetical future election. Probability of voting is a variable measured on a scale from 0 to 10, where 0 means that "I would never vote for this party" and 10 "I would always vote for it". Secondly, through Logistic Regression, we seek to understand the effect that these same independent variables have on vote recall in the Andalusian elections of December 2018. For this, the dependent variable has been dichotomised: those who declared having voted for VOX in the Andalusian elections adopt the value 1, while those indicating other parties are coded with 0. Abstentions, blank or spoiled ballots and DK/NA have been considered lost values in this analysis.

The use of two alternative measures of support for this party allows us to get closer to the elements that explain it, both directly and indirectly, while we also include medium/long-term factors (probability of voting) and short-term factors (vote recall) in the analysis [9]. This is important since the question of "vote recall" is usually biased as a result of various mechanisms such as social desirability or the reconciliation between the memory and the current intention to vote [51], especially if we take into account that it is a party labelled as "radical" [52]. Likewise, the use of two dependent variables capturing electoral support for VOX allows us to have a more robust analysis and counteract the limited size of the group of voters of this party according to their vote recall (n = 60). In the survey, the vote recall for VOX is somewhat lower than the votes obtained by this party (8.7% vs 11.1% of the votes for candidacies respectively), a difference of 2.4 points that is within the margin of error of the survey (±3%). Finally, we selected those covariates from the PACIS survey (2019) that most effectively collected the concepts raised in our hypotheses, as well as those related to other potentially explanatory factors for voting for VOX reviewed in the literature, as control elements. As noted above, having all these variables in a single dataset allows us to check the real weight of each one in the explanation of the VOX vote, something that had not been possible until now due to the limitations

**Table 1. Independent variables included in the regression models.**

| Hypothesis | Variable | Recodification |
|---|---|---|
| **Feminism** | Feminist ideas are fair | In disagreement (1), other categories (0) |
| **Territorial** | Nationalist sentiment | Predominantly Spanish identity (1), other categories (0) |
| | State organisation: Centralisation-Autonomy | Favourable towards greater centralisation of the Autonomous Communities (1), other categories (0) |
| | What to do about Catalonia | "Be more heavy-handed in defending the unity of Spain" (1), other categories (0) |
| **Immigration** | Main problem of Andalusia | Mentions immigration as the first or second problem (1), does not mention immigration as a problem (0) |
| | Building of a mosque in your neighbourhood | In disagreement (1), other categories (0) |
| **Authoritarianism** | Our society does not need tougher government or stricter laws | Disagree (1), other categories (0) |
| | Priority of rights and freedoms in the fight against crime | "Prioritise the fight against crime" (0)–"Prioritise rights and freedoms" (10) (scale) |
| **Populism** | To solve Andalusian problems, people like me would do better than politicians | Agree (1), other categories (0) |
| **Control** | National Government rating (PSOE) | Bad (1), other categories (0) |
| | Andalusia Government rating | Bad (1), other categories (0) |
| | Religious affiliation and practice | *Reference category*: Atheists and non-believers |
| | | *Dummy variables*: Other beliefs; Non-practising Catholics or once a year; Catholics practising monthly or more |
| | Sex | Male (1), Female (0) |
| | Ideological self-placement | "Extreme left" (0)–"Extreme right" (10) (scale) |
| | Age | 18–99 (scale) |
| | Educational level | *Reference category*: Illiterate, without studies and primary level studies |
| | | *Dummy variables*: Secondary or technical studies; University studies |
| | Evaluation of the economic situation of your household over the last 12 months | Average or above average (1), other categories (0) |

Source: Prepared by the authors based on data from the post-electoral PACIS survey (2019)

of the available datasets. These variables are presented in Table 1. For more details on these variables, see Tables A5 and A6 in S1 File.

# Results

Tables 2 and 3 show the results obtained in our analyses. Before computing the complete models, we carried out Linear Regressions to see which group of independent variables fits better into the explanation of the probability of voting for VOX. In this way, we can verify each of the proposed hypotheses in isolation–without including control variables or variables related to other hypotheses–and rank them according to their goodness of fit (Table 2). The set of variables included in each of these regression analyses is specified in Table A5 in S1 File.

According to these measures, the hypothesis that fits better our data is the territorial one (H2.1). Likewise, the weight of other factors such as feminism (H1) or immigration (H2.2) is notable, while authoritarianism (H2.3) lags a bit behind. The populism hypothesis (H2.4) is the one that shows a poorer performance when it comes to explaining the probability of voting for VOX. The detail of the relationship between each covariate and support for VOX is shown in Table 3, where the results of the regression models are collected (Linear for the probability of voting and Logistic for vote recall).

We present here the most parsimonious models after excluding factors that are statistically non-significant in both models and several variables that moderately correlate with other

**Table 2. R² adjusted, AIC and BIC for each model tested by Linear Regression.**

|  | H1 Feminism | H2.1 Territorial | H2.2 Immigration | H2.3 Authoritarianism | H2.4 Populism |
|---|---|---|---|---|---|
| **R² adjusted** | 0.148 | 0.194 | 0.131 | 0.095 | 0.003 |
| **AIC** | 4849.37 | 4616.67 | 4737.23 | 4946.60 | 5140.46 |
| **BIC** | 4868.97 | 4640.99 | 4761.60 | 4971.14 | 5160.16 |
| **N** | 967 | 934 | 946 | 974 | 991 |

Source: Prepared by the authors based on data from the post-electoral PACIS survey (2019)

indicators underlying the same hypothesis (P3 for populism, P24 for immigration, P26 for feminism and P28 for authoritarianism). Including all independent variables in the models or exchanging the items included does not change the results obtained substantially (see Tables A9 and A11 in S1 File).

**Table 3. Results of the regression models of support for VOX.**

|  |  | Probability of vote[a] | Vote recall[b] |
|---|---|---|---|
| **Feminist ideas are unfair** |  | 0.920*** (0.188) | 1.219** (0.392) |
| **Nationalist sentiment** |  | 0.214 (0.266) | 0.171 (0.482) |
| **Recentralisation of the State** |  | 0.811*** (0.185) | 1.117* (0.449) |
| **Heavy-handed approach to defending unity in Spain** |  | 0.632** (0.199) | - 0.654 (0.421) |
| **Immigration as an Andalusian problem** |  | 0.705** (0.237) | 1.467** (0.430) |
| **Rejection of the building of a mosque in your neighbourhood** |  | 0.423* (0.170) | 0.390 (0.426) |
| **Need for a tougher government** |  | 0.607*** (0.171) | 0.581 (0.444) |
| **Fight against crime vs. rights and freedoms** |  | - 0.092** (0.027) | - 0.067 (0.065) |
| **People like me would do better than politicians** |  | 0.034 (0.168) | - 0.100 (0.393) |
| **National Government rating (PSOE)** |  | 0.278 (0.195) | 1.173* (0.507) |
| **Andalusia Government rating** |  | 0.375* (0.185) | 0.857+ (0.503) |
| **Religion (Atheists and Non-believers Cat. Ref.)** | Other beliefs | 0.658 (0.401) | 1.769* (0.834) |
|  | Non-practising Catholics or once a year | - 0.080 (0.195) | - 0.893+ (0.517) |
|  | Catholics practising monthly or more | 0.083 (0.262) | - 0.664 (0.584) |
| **Sex** |  | 0.446** (0.163) | 0.658+ (0.386) |
| **Ideological self-placement** |  | 0.544*** (0.048) | 0.463*** (0.119) |
| **Age** |  | - 0.006 (0.006) | - 0.021 (0.015) |
| **Educational level (Illiterate, Without studies and Primary Studies Cat. Ref.)** | Secondary or technical studies | - 0.325 (0.215) | - 1.070+ (0.568) |
|  | University studies | - 0.140 (0.268) | - 1.286+ (0.665) |
| **Evaluation of the economic situation in your household** |  | 0.303+ (0.167) | 0.424 (0.422) |
|  | ***Constant*** | - 1.792** | - 6.017*** |
|  | ***n*** | 864 | 610 |
|  | ***R²*** | 0.421 | 0.458 |

Source: Prepared by the authors based on data from the post-electoral PACIS survey (2019)

***p < 0.001;

**p < 0.01;

*p < 0.05;

+p < 0.1.

In parentheses the standard errors. For Linear Regression ([a]), the ANOVA test is significant at the level of 0.001; for Logistics ([b]), the Hosmer and Lemeshow test is not significant (> 0.05).

Regarding the Linear Regression model, on the left-hand column of the table, the $R^2$ adjusted of the joint model shows a good fit (0.421). All TOL values are greater than 0.5 and VIF less than 2, the model showing a reduced multicollinearity (for a correlation heat map of covariates and specific TOL and VIF values see Tables A7 to A9 in S1 File). Its results show that most of the variables retained for each hypothesis have a significant effect on the probability of voting for VOX, except for the variable related to populism (H2.4)–result that provides a consistent reading with Table 2 –and for the variable capturing nationalist sentiment (H2.1). Regarding the latter, our data suggests that nationalist sentiment does not seem to be an explanatory factor in voting for VOX *per se*, unlike the desire for greater centralisation of the State of Autonomies–for which previous literature has also shown a positive and statistically significant relationship with the probability of voting for this party [5, 6]–or the will to be heavy-handed towards Catalonia.

Our data shows, however, that voting for this party is much more than a reaction to the territorial issue. Specifically, the hypotheses raised about the influence of anti-feminism (H1), immigration (H2.2), and authoritarianism (H2.3) prove to be significant. If we look at the standardised beta coefficients of the Linear Regression, we see that the ideological self-placement is by far the variable that most contributes to the explanation of the probability of voting for VOX (beta = 0.343)–in line with what had already been pointed out elsewhere [7–9, 11]. After it, the variables that most contribute to the explanation of the probability of voting for VOX are the perception of unfairness in feminist ideas (beta = 0.141), agreement with the recentralisation of the State of Autonomies (beta = 0.131), considering a "tougher" government necessary (beta = 0.101), prioritising the fight against crime over rights and freedoms (beta = -0.094), a heavy-handed approach to defend the unity of Spain (beta = 0,092) and immigration as an Andalusian problem (beta = 0.079). The beta values for each covariate in the Linear Regression are shown in Table A8 in S1 File.

Regarding the control variables–beyond the central role of ideological self-placement–, the gender variable (higher probability of men voting for this party) has proven to be important, as illustrated in previous studies [7, 29]. There is also a relationship between a poor rating of the Andalusian Government and the probability of voting for VOX, but no significant correlation is found for age, educational level, or religion. Finally, we highlight the positive relationship with the interviewee's perception that the economic situation in their household (last 12 months) is "average" or above–significant at the level of 0.1, a weak correlation. Although it is true that no economic hypothesis has been tested here, this finding–however fragile it may be–points in the same direction as the rest of the data obtained so far: those who vote for this party are not below average in economic terms. The income variable was also initially included, but after verifying that its significance is null in both statistical models, we decided to remove it, to increase the n of the final sample, given the significant number of lost cases it includes. Its absence does not modify the adjustment of the models or the conclusions of the results.

Regarding the Logistic Regression model, its overall fit is good, according to the $R^2_{Negelkerke}$ of the model (0.458). In addition to this, its predictive capacity, measured as a global percentage of correct answers, is 91.3% (see Table A10 in S1 File). Looking at the results of this model, which explains vote recall for VOX in the Andalusian elections–the right-hand column of Table 3 –, we see that some of the relationships explained above lose their significance but maintain the same direction in practically all the variables. This may be due to the effects of the low n that we have for this variable: for VOX voters, the effective n in this model is 53. Despite this, a significant relationship reappears in some of the variables indicated so far: ideological self-placement, the unfairness of feminist ideas (H1) or the recentralisation of the State (H2.1) and immigration as the main problem in Andalusia (H2.2). Significant–albeit weak– correlations are also seen in the national Government rating, gender, or educational level, and

for the religious category "other beliefs". Regarding the latter, VOX leadership has met and made public emphasis on his close relationship with evangelical churches, which are the largest religious minority in Andalusia. We have no information about the religious affiliation of our 4 VOX voters belonging to "other religions", but this link is a likely explanation for this result. Finally, no significance was found for nationalist sentiment, neither for the variable on populism, or age.

The result that has changed the most is that of the variable capturing the preference for a heavy-handed approach in Catalonia, even though it is not significant. In any case, both regressions overlap in many issues, with very few dissonances, even though there is an important difference between the two: the variable which focuses on voting probability analyses the group of likely voters (barely 3.2% of lost cases), while the variable which centres on vote recall implies the need to have voted (or, at least, claim to have done so). This generates unequal samples, not only in their size but also in their composition, leaving abstentions and blank or spoiled ballots out of this comparison, thus potentially generating a level of bias [40].

As additional robustness checks we performed Logistic Regressions comparing VOX voters with those of the other two centre-right parties (see Tables A12 and A13 in S1 File). The results of both regressions are coherent with the arguments presented so far, with feminism (and gender), as well as immigration as MIP being significant in both cases. Other variables do not reach statistical significance, like ideological differences in the case of VOX vs PP or nationalist feelings for VOX vs PP/Ciudadanos.

## Discussion and conclusions

Previous analyses on voting for VOX in Spain have made an important contribution, showing the centrality of the territorial conflict in explaining the electoral surge of this party. Our main argument is that it is not possible to understand this vote without also considering complementary explanations. Particularly, our central contribution lies in showing that the reaction against feminist ideas is one of the central explanatory factors of the VOX vote since its inception, something that had not been sufficiently explored in previous research. Also, having a questionnaire specifically designed to understand the VOX vote, we simultaneously tested alternative hypotheses that, until now, had not been duly verified empirically, such as authoritarian attitudes. The full explanatory model shows that both the rejection of different expressions of a more diverse and egalitarian society (feminism and immigration) as well as authoritarianism and ideology are essential components of support for this party, together with the territorial issue.

This multi-causal explanation makes the electoral surge of VOX a less exceptional phenomenon in the field of *European Radical Right Studies* [15]. Although at European level research has shown considerable diversity in the causes of voting for the radical right in each country, VOX's emergence appeared to be a somewhat unusual phenomenon. The territorial component was identified as the main explanatory factor for its rise while other factors, central to many other Western European countries, were absent. Our analysis, based on the first electoral success of the party shows that, from the first moment, this success was based on a combination of explanations similar to those of other Western European radical right parties.

The variables associated with the territorial conflict have proven to fit well into the explanation of voting for VOX, playing an important role on it even when we control for the main alternative explanations. This reinforces the certainty about the importance of this factor, which has been an important argument for VOX [53], but also for previous parties (UPyD or Ciudadanos) that had already emerged to represent the new recentralising demand of substantial parts of Spanish society [30]. Indeed, our work suggests that preferences regarding the

territorial structure of the State are the most important part of the story, rather than the intensity of nationalist sentiment, or even specific preferences regarding the Catalan conflict. Thus, VOX's defence of "national unity" would be behind its success [31]. And, although the Catalan conflict triggered emotions and attention when this election was held, territorial tensions go far beyond it; hence, this factor may still be present despite the disappearance of the Catalan conflict from the centre of the political agenda. The specific role of each of these ideas should be the subject of future analysis.

However, according to our data, the reaction against the values of the "libertarian left"–particularly against feminism–seems to play an equally important role in VOX's electoral surge. Although this hypothesis has been illustrated in previous work [17, 18], until now there was no sufficient empirical evidence of its explanatory role. If our interpretation is correct (that feminism is the target of their attacks as the most successful force among the set of values they oppose) it means that, had VOX emerged electorally around 2004 –the year homosexual marriage was approved in Spain–, perhaps its greatest outside enemy would have been the movement aimed at expanding the rights of the homosexual community. In fact, the two previous tests of the counter-libertarian hypotheses had used variables related to the Gay's rights community [7, 13]. Thus, feminism is now their preferred target precisely because of its salience and its capacity for mobilisation in recent years. This is similar to what Mendes and Dennison [14] state for the immigration issue. They also point the salience of anti-feminism, an issue that VOX successfully mobilised on [14: 769]. However, the only aspect that our data allows us to affirm with certainty is that the rejection of feminist ideas has been an important differential factor in support for VOX during this period, precisely because this party has encouraged this rejection with much more clarity than other forces from the traditional right.

A rejection of immigration has appeared also as a relevant factor in all our analyses, and this has occurred despite anti-immigration attitudes holding only moderate weight in Spanish society [12]. This relevance places Spanish results in line with those of other Western European countries, where this rejection is an important factor in many cases [3]. Further research should unravel the extent to which these kinds of attitudes are part of a "volatile scepticism" about immigration or not [32].

The last of the verified hypotheses is the one with the least quantitative importance, yet it is probably one of the most novel, as it does not have almost any prior empirical verification for the Spanish case: the weight of support for authoritarian values and tough security policies as a priority. Given the potential importance of this explanation in countries with a recent authoritarian past, where ambiguous assessments of the legacy of the dictatorship are abundant and more extended among right-wing voters [38, 54], it would be important to continue to explore this idea and the role for security issues and punitive culture.

The hypothesis that VOX was an embodiment of the populist protest is not supported in this study. As previous research has shown [13, 16], populist slogans are not central to the discourse of this party and, if they do appear, they are always beneath those of nationalist rhetoric. Thus, even if many of the most well-known comparative datasets continue to consider VOX as a populist party, we did not find evidence in this direction. It is true, in any case, that we have been able to verify only one aspect of a multidimensional concept such as populism, so it is necessary to have additional indicators to further analyse this possibility. The fact that those who voted for VOX differ from others by being the most categorically critical with the previous Andalusian Government–even controlling for ideology and all the other variables–could constitute a favourable indication towards thinking that this party does display some type of protest or visceral rejection of many government actions.

This result is clearly connected to the role that ideology plays. According to our results, this variable most effectively explains the vote for this party in strictly empirical terms. Therefore,

it does not make sense to present VOX as a party that fosters discontent in a transversal way in the populist style. Even if the centrality of ideology in explaining voting for any party is a constant in Spanish electoral behaviour [48] and not a singular factor of this party, it is a fundamental piece of information for identifying the particularities of the Spanish case (where the appearance of former left voters among VOX voters is quite rare) with respect to the radical right at European level.

Finally, and although no economic explanation as such has been tested in this study, the control variable relative to the interviewees' evaluation of their economic situation does suggest that VOX voters are not characterised by economic hardship. It is, in fact, the opposite. For the Spanish case, and in line with what Engler and Weisstanner [47] illustrate, those who perceive that have the most to lose are those who choose to vote for the radical right.

From a methodological point of view, the double approach to the subject through the questions of vote recall and voting probability gives rise to more robust results. Both approaches share most results and point to a confirmation of most of our hypotheses and a rejection of the populist hypothesis. The small differences that appear–mainly related to variables which are or are not significant in one model or another–may be related to the limitations arising from the small number of actual VOX voters in the sample. Also, with the real differences that exist between choosing to vote for this party in a specific election (vote recall) or being predisposed to do so in general (probability scale) [40]. This final difference will need further investigation in order to understand the lower and upper limits of support for this political party.

Three final remarks and implications are needed. First, regarding the generalizability of the results, the Andalusian elections of 2018 represented the electoral surge of VOX. As such, they represent a great opportunity to understand how their electorate was formed for the first time. Our analysis has shown that, from the very first minute, their success was based on quite more than territorial politics. Some of their defining characteristics at that time have probably changed today in Andalusia or Spain. However, the similarities shown through the text with the situation of many other European radical right parties, as well as VOX's relative electoral stability since then [7], suggests that there are not so many singularities in this 2018 electoral process, and that there are many anticipatory signs of what VOX continues to represent in Spain and in comparative perspective. For example, regarding our main hypothesis, a recent survey [55] shows that two anti-feminist proposals (limiting abortion rights and abolish the Law against gender-based violence) are the second and third VOX proposals more easily identified by the Spanish population, thus showing that this idea continues to be a central part of the party's identity.

Second, the results showing the anti-feminism of VOX voters and the diversity of results– and measures–that exists on this relationship in comparative perspective [19] should contribute to raise questions about the exact content and role that the anti-libertarian agenda plays in the explanation of support for radical right parties. This could include the relationship between gender and immigration agendas [56] or how both cultural and socio-economic issues exactly contribute to the growth of this party family [57]. For example, is anti-feminism the specific format that the anti-libertarian reaction takes when feminism is powerful (as we suggest in the text) or is anti-feminism another issue that "challenger parties" [58] can use to show their anti-mainstream character?

Finally, the relevance of these findings goes well beyond the field of electoral politics. While the presence of radical right parties in government has still been limited, their policy influence exists both through the incorporation of their ideas on platforms of other parties, as well as through their influence in the formation of parliamentary majorities and voting of Laws and public budgets. Andalusia was the first scenario where VOX were able to exert this influence– even if it did so from outside government–and one of their main priorities and outcomes has

been precisely on gender equality policies [59], thus showing policy effects resulting from their anti-feminist orientation.

## Supporting information

**S1 File. Methodological appendix.**
(DOCX)

## Acknowledgments

The authors would like to thank those who made the development of the PACIS (2019) survey possible, especially Sebastian Rinken, Manolo Trujillo and all the UTEA staff involved.

## Author Contributions

**Conceptualization:** Joan Font.

**Data curation:** Sara Pasadas-del-Amo.

**Formal analysis:** Rodrigo Ramis-Moyano, Sara Pasadas-del-Amo, Joan Font.

**Investigation:** Rodrigo Ramis-Moyano, Sara Pasadas-del-Amo, Joan Font.

**Methodology:** Rodrigo Ramis-Moyano, Sara Pasadas-del-Amo.

**Supervision:** Rodrigo Ramis-Moyano, Sara Pasadas-del-Amo, Joan Font.

**Writing – original draft:** Rodrigo Ramis-Moyano, Sara Pasadas-del-Amo, Joan Font.

**Writing – review & editing:** Rodrigo Ramis-Moyano, Sara Pasadas-del-Amo, Joan Font.

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
