## [Decision Letter · Decision Letter 0]

8 Nov 2022

PONE-D-22-22220

NOT ONLY A TERRITORIAL MATTER: THE ELECTORAL SURGE OF VOX AND THE ANTI-LIBERTARIAN REACTION

PLOS ONE

Dear Dr. Ramis Moyano,

Thank you for submitting your manuscript to PLOS ONE. After careful consideration, we feel that it has merit but does not fully meet PLOS ONE’s publication criteria as it currently stands. Therefore, we invite you to submit a revised version of the manuscript that addresses the points raised during the review process.

R1 provides very useful comments and suggestions. I believe that you can address them all. 

R2 is more critical. While PLOS One is explicit about its criteria for the publication and that the ‘scope of the contribution’ is not among them, R2’s comment about the clarity of the contribution is very useful. Please clarify the contribution of the manuscript and its implications keeping in mind the literature and R2’s comments. The rest of R2’s comments are also useful and I believe that you can address them all in revising the manuscript.

We look forward to receiving your revised manuscript.

Kind regards,

Jean-François Daoust

Academic Editor

PLOS ONE

Journal Requirements:

Reviewers' comments:

Reviewer's Responses to Questions

**Comments to the Author**

1. Is the manuscript technically sound, and do the data support the conclusions?

Reviewer #1: Yes

Reviewer #2: Partly

2. Has the statistical analysis been performed appropriately and rigorously? 

Reviewer #1: Yes

Reviewer #2: Yes

3. Have the authors made all data underlying the findings in their manuscript fully available?

Reviewer #1: Yes

Reviewer #2: No

4. Is the manuscript presented in an intelligible fashion and written in standard English?

Reviewer #1: Yes

Reviewer #2: Yes

5. Review Comments to the Author

Reviewer #1: Thank you for giving me the possibility to review your manuscript “Not only a territorial matter: The electoral surge of VOX and the anti-libertarian reaction”. The manuscript is really well written, easy to understand and transparent. The results section is carefully written and takes, from my perspective, most relevant issues into account. I would encourage the authors to address the following issues regarding their methodological part before re-submission to PLOS ONE.

The authors report that co-linearity is not an issue for their study. However, I still wonder to what extent single variables of interest correlate with each other. This is mostly because I believe that some of the questions do not only tap into their foreseen theoretical concept, but potentially grasp parts of the variation due to other concepts. Most importantly, left-right self-placement and attitudes towards feminism, authoritarian attitudes and nationalism (especially the item “state organization” calling for harsher action against regional dissenters) and immigration and left-right self-placements might not be measuring completely different phenomena. It would be great if the authors could show a co-linearity plot in the appendix in addition to the VIF and TOL reported to give readers an intuitive inside into potential issues with co-linearity.

I wonder, why are not all variables included in the final model that are listed in Table A5? The authors state in a note to Table A5 that they have not included all variables into the main models. Does it change the results if they do?

This might be especially interesting for the populism item. Is your finding robust to using the other item that populism does not seem to explain (much) of the variance in support for VOX? I would not be surprised, and the authors acknowledge that their test of populism is not a particularly rigorous one focusing on only one question. Taking the second item into account might be a stronger test.

It would also be great if the authors could add a short codebook to the supplementary information so that reviewers (and others) can replicate their findings more easily. As of now, it is sometimes difficult to tell which variables link to which item in the study, even although I can read Spanish.

There are some typos that the authors might want to take care of:

p.8 “Marcos-Marne et al 2021)suggests”

p.11 “gained parliamentary representation in Spain since [the] 1979 general election”

p.13 “within the margin of error of the survey (+-3%).Finally”

Reviewer #2: Review for “NOT ONLY A TERRITORIAL MATTER: THE ELECTORAL SURGE OF VOX AND THE ANTI-LIBERTARIAN REACTION”

This is an interesting paper. It seeks to add to the, now quite rich, literature on the electoral determinants of VOX, Spain’s (relatively) new radical right-wing party. The paper’s central claim, as currently framed, is that there is more to the story than just concerns over the territorial conflict in Catalonia.

Whilst I find the paper, on the whole, interesting and methodologically straightforward, I am not entirely convinced that the paper currently makes enough of an independent contribution to merit publication. There is scope for a more concise and theoretically streamlined iteration of the paper to perhaps be published as a research note. I detail my comments below. Whilst somewhat critical of the paper in its current form, I do hope that they are useful for the authors in terms of revising their paper.

Main concerns

1. My primary concern is related to the paper’s claim to a contribution. The paper begins by signalling that there is more to the story than the Catalan crisis and that these factors need to be considered too. The paper then goes on, however, to highlight the vast catalogue of work that highlights the different factors that scholars have considered. The paper mentions the work by Marcos-Marne an colleagues published in Politics. This paper looks at the issue of gender and women’s issues, theorising and demonstrating it does indeed play a role. Given this extant literature, what is the paper’s central novel thesis and/or finding? If it is that views on the perceived “fairness” of feminism is distinct from views on women’s issues, then this may well be more clearly argued.

2. I find the somewhat summative (and long) list of hypotheses not necessarily helpful for the paper. Is this a paper of “who votes for VOX?” – that is how the long list of hypotheses currently reads and as mentioned, that seems well answered. My recommendation would be focus on the main explanatory variable of the paper and make this more a independent variable focused paper that hones in on what the author’s identify as being important and make a strong case for why it should be considered. Of course, authors write papers and not reviewers, but I provide this recommendation in order to help the authors find a mechanism via which they can more clearly sell what the value-added component of their contribution is. Essentially the summative hypotheses are just controls for your main variables of interest.

3. It would be useful for the authors to make more of a case for the external validity of the case they consider. Andalucian elections in 2018 feel very remote in time from this vantage point, we have had numerous national, supranational, regional, and municipal elections since then and VOX has had varying levels of success across these elections. Can the claims of what mattered in region X and time t-x wield external validity now? I don’t necessarily think that they don’t, but I do have some concerns about this.

4. Finally, I have concerns of the substantive interpretation of the findings in relation to the broader claims of “why” VOX was successful. The paper effectively demonstrates that anti-feminist believes wield a sizeable and significant independent effect on support for the party. Is that equitable to the events of Catalonia. In a counterfactual scenario where the events of the Catalan crisis didn’t occur, is it the paper’s claim that anti-feminists’ positions would have been enough to explain VOX’s success? This is somewhat implicit throughout the paper.

Minor points

1. The discussion of the different r2 between different models was a bit unusual and I’m not convinced the interpretation that accompanies this comparison is doing what the paper’s authors believe it is doing.

6. PLOS authors have the option to publish the peer review history of their article (what does this mean?). If published, this will include your full peer review and any attached files.

Reviewer #1: No

Reviewer #2: No

---

## [Author Response · Author response to Decision Letter 0]

14 Dec 2022

Dear PLoS ONE Editor and Reviewers,

Thank you very much for taking the time to review our work and give us your valuable feedback. We have edited the manuscript to address your concerns and we feel that the outcome has improved as a result of this revision. We answer to each of your comments below and describe the relevant changes that we have made to the manuscript.

Editorial Comment:

Answer and Revisions:

We have completely revised our files to match PONE style and format guidelines.

Editorial Comment:

Answer and Revisions:

We have checked the grant information and corrected it when resubmitting the files.

Editorial Comment:

Answer and Revisions:

We have requested to remove this option.

Editorial Comment:

4. Please include captions for your Supporting Information files at the end of your manuscript, and update any in-text citations to match accordingly.

Answer and Revisions:

Captions for Supporting Information appendix are now included at the end of the main manuscript text. In-text citations now refer to specific supporting tables, where appropriate.

Reviewer's Questions:

1. Is the manuscript technically sound, and do the data support the conclusions?

Reviewer #1: Yes

Reviewer #2: Partly

Response and Revisions:

Following your suggestions, we have rechecked our statistical analyses and added the regression models including all our independent variables as supporting information (Tables A9 and A11 in S1 Methodological Appendix). We have also computed a correlation heatmap (Table A7 in S1 Methodological Appendix) and made transparent in the main text our model specification (lines 341 to 346, p. 17 in the unmarked manuscript). We have modified the description of table 2 in order to better explain the use of r2 adjusted to rank alternative hypotheses according to their explanatory power (lines 317 to 330, p.15) and clarified the links between results and conclusions throughout the main text. We feel that the paper has improved in transparency, robustness and clarity as a result of this process.

Reviewer's Questions:

3. Have the authors made all data underlying the findings in their manuscript fully available? 

Reviewer #1: Yes

Reviewer #2: No 

Response and Revisions:

Together with our initial submission, we uploaded an anonymized data set as Supporting Information that allowed replication of our analyses. However, since then we have also published the complete underlying dataset and documentation of this survey, that is now open-access and available for download here: (https://doi.org/10.20350/digitalCSIC/14804).

Reviewer comments:

Reviewer #1:

The authors report that co-linearity is not an issue for their study. However, I still wonder to what extent single variables of interest correlate with each other. This is mostly because I believe that some of the questions do not only tap into their foreseen theoretical concept, but potentially grasp parts of the variation due to other concepts. Most importantly, left-right self-placement and attitudes towards feminism, authoritarian attitudes and nationalism (especially the item “state organization” calling for harsher action against regional dissenters) and immigration and left-right self-placements might not be measuring completely different phenomena. It would be great if the authors could show a co-linearity plot in the appendix in addition to the VIF and TOL reported to give readers an intuitive inside into potential issues with co-linearity.

Response and Revisions:

We have added a correlation heatmap of all covariates as table A7 in S1 Methodological Appendix. Several variables that moderately correlate (0.3-0.4) with others indicators underlying the same hypothesis or dimension (P3 for populism, P24 for immigration, P26 for feminism, P28 for authoritarianism or household income) had already been excluded from our final models. Overall, discounting correlation between dummy variables computed from the same categorical variable (i.e. education level, NEST-R, or religious practice, PRELIG), the heatmap portraits a similar scenario as co-linearity measures (VIF and TOL) that we presented in the models. All of them show that co-linearity is not affecting our models in a significant manner. Even if ideology, as you pointed out, is somehow correlated to many of our attitudinal variables, the strength of this association is limited (the highest ones being around 0.3).

Reviewer #1:

I wonder, why are not all variables included in the final model that are listed in Table A5? The authors state in a note to Table A5 that they have not included all variables into the main models. Does it change the results if they do?

This might be especially interesting for the populism item. Is your finding robust to using the other item that populism does not seem to explain (much) of the variance in support for VOX? I would not be surprised, and the authors acknowledge that their test of populism is not a particularly rigorous one focusing on only one question. Taking the second item into account might be a stronger test.

Response and Revisions:

We have included the full models as Tables A9 and A11 in S1 Methodological Appendix. As you can see, including all variables do not alter our results significantly. Moreover, the loss of explanatory power in the final model as compared to the complete model is almost negligible for the linear regression (0.427 to 0.421), and limited for the logistic regression (0.512 to 0.458, global percentage of classification from 92.4% to 91.3%). We have also computed the models exchanging the items excluded in each dimension and this does not improve them or produce relevant changes in the results. We do not show those additional models in the appendix for the sake of simplicity.

Reviewer #1:

It would also be great if the authors could add a short codebook to the supplementary information so that reviewers (and others) can replicate their findings more easily. As of now, it is sometimes difficult to tell which variables link to which item in the study, even although I can read Spanish.

Response and Revisions:

The complete dataset of the survey, together with codebooks in Spanish and English, have been published in the open access institutional repository of the Spanish National Research Council, that follows best practices in data sharing and management. All the information regarding this survey dataset can be accessed here: https://doi.org/10.20350/digitalCSIC/14804 . We have added the variable labels in the original dataset to table A5 in the Supplementary Information to ease replication of the analyses and findings discussed in the manuscript.

Reviewer #1:

There are some typos that the authors might want to take care of:

p.8 “Marcos-Marne et al 2021)suggests”

p.11 “gained parliamentary representation in Spain since [the] 1979 general election”

p.13 “within the margin of error of the survey (+-3%).Finally”

Response and Revisions:

Thank you very much for such a detailed and careful revision. We have corrected these typos in the new version.

Reviewer #2:

Whilst I find the paper, on the whole, interesting and methodologically straightforward, I am not entirely convinced that the paper currently makes enough of an independent contribution to merit publication. There is scope for a more concise and theoretically streamlined iteration of the paper to perhaps be published as a research note.

1. My primary concern is related to the paper’s claim to a contribution. The paper begins by signalling that there is more to the story than the Catalan crisis and that these factors need to be considered too. The paper then goes on, however, to highlight the vast catalogue of work that highlights the different factors that scholars have considered.

Response and Revisions:

Perceived novelty of research results is not a relevant publication criterion for PLoS journals. However, we do think that our paper indeed makes relevant and novel contributions to this area of research. Following your suggestion, we have clarified them in the manuscript (lines 65 to 79, pages 3 and 4) and justify further in the lines below.

We depart from an original dataset that is the outcome of a survey that we specifically designed to test if the main explanatory factors for the rise of other Western European radical right-wing parties applied to the election where Vox got represented in a parliament for the first time in Spain (the 2018 Andalusian regional election). Our results – using two alternative measures of support for VOX – show that the preferences and attitudes related to the territorial conflict do matter, but their relevance is clearly reduced when we add alternative explanations.

The main contribution of our work is twofold. First, our analysis provides empirical evidence on the importance that several factors, for which there are limited data in Spain, had to explain VOX performance in that particular election. These factors are authoritarian attitudes and opposition to feminism, with the latter showing an important explanatory power of support for VOX, even after controlling for other alternative factors. Second, by being able to model the different factors in the same dataset, we can rank them according to their relevance explaining the electoral surge of this party. This is something that can’t be done using secondary data, such as CIS post-election surveys or political barometers, because they lack variables that would contribute to build a more complete explanation of the vote for individual parties, particularly the most recent ones such as VOX.

Different factors explaining “who votes for Vox” have been considered in the literature using different sources of data regarding different elections, but most of them have based their analyses in secondary data that lacked information about some of these explanations (Turnbull-Dugarte 2019; Ortiz 2019: Turnbull-Dugarte et al. 2020). Other commercial surveys have tackled a wider array of factors, but their results have been reported in the daily press in a necessarily simple and descriptive way (El País, 2018). We have tried to be really careful by reviewing and giving credit to all those works in our manuscript.

Regarding anti-feminism, most previous research has approached this subject from the supply side. Some of this research hypothesises on the effects that the anti-feminism in the party manifesto and leaders’ discourse may have in the observed gender gap in support for this party (Rama et al. 2021: page 71; Turnbull-Dugarte, 2019; Mendes and Dennison, 2021). Other authors have described and analysed the anti-feminist discourse produced by this party, via quantitative and content analysis of the party’s manifestos (Cabeza et al. 2021; Fernández Suárez, 2021), parliamentary speeches (Fernández Suárez, 2021), publications in social media (Bernárdez- Rodal et al. 2020, Luque and Cano, 2020) or all of them (Álvarez Benavides and Jiménez Aguilar, 2021).

From the demand side, and under the same hypothesis used here of an anti-libertarian reaction, Rama et al. 2021, analysing European Social Survey data, show the link between support for Vox and opposition to LGBT+ rights, specifically opposing same-sex marriage.

As far as we are aware, our paper is the first academic analysis that empirically proves the link between having anti-feminist attitudes and voting for Vox. Moreover, our data allow to control for other alternative explanations and rank the importance of all these factors according to their explanatory power.

Reviewer #2:

The paper mentions the work by Marcos-Marne an colleagues published in Politics. This paper looks at the issue of gender and women’s issues, theorising and demonstrating it does indeed play a role. Given this extant literature, what is the paper’s central novel thesis and/or finding? If it is that views on the perceived “fairness” of feminism is distinct from views on women’s issues, then this may well be more clearly argued.

Response and Revisions:

The paper published by Marcos Marne et al. in Politics in 2021 does not deal with gender and women issues. The paper considers whether demand-side populism plays a role in VOX’s vote share. As they put it: They “combine holistic grading of key speeches and electoral manifestos, with the analysis of innovative survey data to respond to two main research questions: are populist ideas central to the electoral discourse of VOX; and is VOX more electorally attractive to voters who themselves display stronger populist attitudes?”. 

In this paper, preferences that tap into lesbian, gay, bisexual, transgender (LGBT) and women’s rights are included as a control in the regression models together as a “new variable that represent the latent structure of preferences of the individuals on the topic” and are in fact statistically non-significant in both models, computed to explain the declared probability of voting for VOX and VOX voting intention (see model 5 column in tables 3 and 4 of the paper, pages 9 and 10). Thus, that paper makes a relevant contribution, but a quite different one from our text, which empirically proves the link between having anti-feminist attitudes and voting for Vox.

Reviewer #2:

2. I find the somewhat summative (and long) list of hypotheses not necessarily helpful for the paper. Is this a paper of “who votes for VOX?” – that is how the long list of hypotheses currently reads and as mentioned, that seems well answered. My recommendation would be focus on the main explanatory variable of the paper and make this more a independent variable focused paper that hones in on what the author’s identify as being important and make a strong case for why it should be considered. Of course, authors write papers and not reviewers, but I provide this recommendation in order to help the authors find a mechanism via which they can more clearly sell what the value-added component of their contribution is. Essentially the summative hypotheses are just controls for your main variables of interest.

Response and Revisions:

Yes, this is a paper that intends to answer who voted for Vox in the election where this party got represented in a parliament for the first time in Spain (the 2018 Andalusian regional election). Published articles on this issue (Turnbull-Dugarte, 2019 and Ortiz, 2019) employed data from CIS post-election survey, a high-quality source of data, that follows a standard questionnaire but did not include variables allowing to test factors that have proved important when explaining the emergence of radical right-wing parties in Europe. Our data, coming from a different high-quality post-election survey that was designed to explain ad-hoc the results of that election, allow us to show a more complete picture of the factors that played a role on that occasion and to rank them according to their explanatory power. When presenting our findings, we have underlined the factor that was most novel in this research area, anti-feminist attitudes, but treating the rest of factors as mere controls would imply, in our opinion, loosing valuable insights on the subject. We appreciate the recommendation, but consider that the current structure reflects better the goals and contributions of the paper.

Reviewer #2:

3. It would be useful for the authors to make more of a case for the external validity of the case they consider. Andalucian elections in 2018 feel very remote in time from this vantage point, we have had numerous national, supranational, regional, and municipal elections since then and VOX has had varying levels of success across these elections. Can the claims of what mattered in region X and time t-x wield external validity now? I don’t necessarily think that they don’t, but I do have some concerns about this.

Response and Revisions:

We answer to the question of generalizability and external validity in lines 523 to 536 (pages 24 and 25) of the manuscript:

“First, regarding the generalizability of the results, the Andalusian elections of 2018 represented the electoral surge of VOX. As such, they represent a great opportunity to understand how their electorate was formed for the first time. Our analysis has shown that, from the very first minute, their success was based on quite more than territorial politics. Some of their defining characteristics at that time have probably changed today in Andalusia or Spain. However, the similarities shown through the text with the situation of many other European radical right parties, as well as VOX’s relative electoral stability since then (Rama et al 2021), suggests that there are not so many singularities in this 2018 electoral process, and that there are many anticipatory signs of what VOX continues to represent in Spain and in comparative perspective. For example, regarding our main hypothesis, a recent survey (El País, 2022) shows that two anti-feminist proposals (limiting abortion rights and abolish the Law against gender-based violence) are the second and third VOX proposals more easily identified by the Spanish population, thus showing that this idea continues to be a central part of the party’s identity”.

Reviewer #2:

4. Finally, I have concerns of the substantive interpretation of the findings in relation to the broader claims of “why” VOX was successful. The paper effectively demonstrates that anti-feminist believes wield a sizeable and significant independent effect on support for the party. Is that equitable to the events of Catalonia. In a counterfactual scenario where the events of the Catalan crisis didn’t occur, is it the paper’s claim that anti-feminists’ positions would have been enough to explain VOX’s success? This is somewhat implicit throughout the paper.

Response and Revisions:

This is an interesting counterfactual question. The truth is that we don’t know if VOX mobilization of anti-feminist positions would have been enough to be present in the Andalusian parliament after the 2018 regional election. All that we can safely say looking at our data is that the emergence of VOX in Andalusia can be explained as a consequence of multiple factors where the variables associated with the territorial conflict were the most important, even after controlling for the main alternative explanations and that the reaction against the values of the “libertarian left”, particularly against feminism but also immigration, also played a relevant role. Our data allow us to explicitly rank the importance of the different factors contributing to explain the vote for Vox in the 2018 Andalusian election, which we do in the Discussion and conclusions section.

Reviewer #2:

1. The discussion of the different r2 between different models was a bit unusual and I’m not convinced the interpretation that accompanies this comparison is doing what the paper’s authors believe it is doing.

Response and Revisions:

We have modified the description of table 2 and rewritten that section in order to better explain the use of R2 adjusted to rank alternative hypotheses according to their explanatory power (lines 317 to 330, p.15).

Thanks again to all of you for what we feel has been a productive review process.

---

## [Decision Letter · Decision Letter 1]

5 Feb 2023

PONE-D-22-22220R1NOT ONLY A TERRITORIAL MATTER: THE ELECTORAL SURGE OF VOX AND THE ANTI-LIBERTARIAN REACTIONPLOS ONE

Dear Dr. Ramis Moyano,

Thank you for submitting your manuscript to PLOS ONE. After careful consideration, we feel that it has merit but does not fully meet PLOS ONE’s publication criteria as it currently stands. Therefore, we invite you to submit a revised version of the manuscript that addresses the points raised during the review process. R1's suggestions are useful and would improve the manuscript. Please revise your manuscript based on their comments. When I will receive the new iteration, I will not send it back to the reviewers and will have a final reading before taking a final decision.   Please submit your revised manuscript by Mar 22 2023 11:59PM. If you will need more time than this to complete your revisions, please reply to this message or contact the journal office at plosone@plos.org. Please include the following items when submitting your revised manuscript:A rebuttal letter that responds to each point raised by the academic editor and reviewer(s). You should upload this letter as a separate file labeled 'Response to Reviewers'.A marked-up copy of your manuscript that highlights changes made to the original version. You should upload this as a separate file labeled 'Revised Manuscript with Track Changes'.An unmarked version of your revised paper without tracked changes. You should upload this as a separate file labeled 'Manuscript'.If applicable, we recommend that you deposit your laboratory protocols in protocols.io to enhance the reproducibility of your results. Protocols.io assigns your protocol its own identifier (DOI) so that it can be cited independently in the future. For instructions see: https://journals.plos.org/plosone/s/submission-guidelines#loc-laboratory-protocols. Additionally, PLOS ONE offers an option for publishing peer-reviewed Lab Protocol articles, which describe protocols hosted on protocols.io. Read more information on sharing protocols at https://plos.org/protocols?utm_medium=editorial-email&utm_source=authorletters&utm_campaign=protocols.

We look forward to receiving your revised manuscript.

Kind regards,

Jean-François Daoust

Academic Editor

PLOS ONE

Journal Requirements:

Reviewers' comments:

Reviewer's Responses to Questions

**Comments to the Author**

1. If the authors have adequately addressed your comments raised in a previous round of review and you feel that this manuscript is now acceptable for publication, you may indicate that here to bypass the “Comments to the Author” section, enter your conflict of interest statement in the “Confidential to Editor” section, and submit your "Accept" recommendation.

Reviewer #1: All comments have been addressed

Reviewer #2: All comments have been addressed

2. Is the manuscript technically sound, and do the data support the conclusions?

Reviewer #1: Yes

Reviewer #2: Yes

3. Has the statistical analysis been performed appropriately and rigorously? 

Reviewer #1: No

Reviewer #2: Yes

4. Have the authors made all data underlying the findings in their manuscript fully available?

Reviewer #1: Yes

Reviewer #2: Yes

5. Is the manuscript presented in an intelligible fashion and written in standard English?

Reviewer #1: No

Reviewer #2: Yes

6. Review Comments to the Author

Reviewer #1: First and foremost, I want to re-iterate that I enjoyed reading the manuscript, and that I think the authors have made great progress on the cohesiveness and transparency of their manuscript. Their study sheds great and detailed light on parties others need to categorized for large N comparative studies.

From my perspectives, the authors have responded sufficiently to the reviewers’ comments. I noticed two minor things regarding the wording which the authors could clarify for their readers, some typos and one larger point regarding their discussion of R² values. I would like to ask the authors to address these minor issues before publication.

Minor things regarding wording

I find the sentence “all the covariates included in the remaining hypotheses…” (p. 17, line 355) confusing, because there has been no hypothesis discussed briefly before and I struggle to understand which hypotheses, if not all, the authors refer to exactly.

Some of the conclusions are not fully clear to me. In your discussion, you state that “the variables associated with the territorial conflict are those which contribute most to explaining the vote for VOX” (p. 21, line 445-6), but I do not see that reflected in Table 2 nor in the beta coefficients presented in Table A9. Maybe you could rephrase this sentence to reflect that, e.g. anti-feminist attitudes, seem to play an at least equally important role – which I assume was also one of the main arguments you wanted to make. As of know, it reads as a general statement that all else being equal, positions regarding the Spanish territorial conflicts are most important. Similarly, the discussion states that “a rejection of immigration has been the third most important factor” (p. 22, line 475), although this is not reflected in the beta coefficients in Table A9. More than two factors show a larger beta coefficient, e.g. the fight against crime.

Discussion of R² values

I would like to rise one issue that reviewer 2 had previously mentioned as a minor point. Your response and adjustments point out that you want to rank models in Table 2 based on their explanatory power using the adjusted R². From what I learned, the adjusted R², in contrast to the R², cannot be (easily) interpreted as the percentage of variance explained by the model because it does not only take variance into account, but also the number of variables included in the model. While thinking about possible solutions easy to implement during a second round of reviewing, other questions arose:

What are the underlying models for Table 2? Most importantly: which set of variables do they contain? Do they contain control variables or only variables for each hypothesis (feminism, territorial, …)? For each hypothesis, do the models contain the full set of items (Table A5) or only the item included in the regression analyses in Table 3?

Depending on the answers to these questions, the authors might consider 1) using the simple R² and compare the models based on that (if the models contain the same number of variables, and if the set of respondents would be the same in each model), 2) just describing the simple R² as percentage of variance explained but refrain from comparing models, or 3) tone down the language on the percentage of variance explained by the model. If you want to stick to the model comparison, I would recommend to use additional criteria such as AIC and BIC to compare the models with regard to their goodness of fit because the models are not nested nor do they seem to contain the same number of variables.

I can recommend A Guide to Modern Econometrics by Marno Verbeek for these things, which I regularly use myself, especially pages R², AIC, BIC and model comparisons.

I do not think that the model comparison and Table 2 is super important to answer your research question, which is why I would be ok with what-ever solution you find. I would just like to encourage you to be more precise in describing and interpreting the models presented in Table 2 or alternative models you might choose. It would also be great if the authors could add the information regarding the questions above to the supplementary material.

Similarly, I would like to ask the authors to reconsider their interpretation of the Negelkerke R². I have never used it before, but as far as I understood, this measure, again is not based on the variance (only) and thus, should not be interpreted as the percentage of the variance explained by the model. Again, one solution might be to simply delete the statement that this is equal to “explaining 46% of the variability of vote recall” (p. 19, line 389-90) and simply point out that the model fit seems great.

Typos

p. 2, line 37 “these voters [have not been] so [different] from voters of other European ...”

p. 17, line 342-3: “that moderately correlate with otherS indicators”

p. 17, line 345: “does not change [] the results [substantially]”? (English not my mother tongue, so maybe I am wrong here.)

p. 22, line 479: “feminism, [an] issue that VOX …”

p. 19, line 410: “the variable which focus[es]...”

In Table A9, A11 and A12 of the appendix, the authors use the “,” instead of the “.” for their decimals (e.g. 0,877 instead of 0.877).

Reviewer #2: Second review for “NOT ONLY A TERRITORIAL MATTER: THE ELECTORAL SURGE OF VOX AND

THE ANTI-LIBERTARIAN REACTION”

The authors have made some updates to their initial submission, largely in response to some of the concerns raised by reviewer 1.

My initial primary concern with the paper was that the article did not make any original contribution to knowledge based on empirical findings or on theoretical arguments. I made some suggestions about changing the focus of the paper to make it more theoretically rich as opposed to empirically descriptive, but the authors have opted to retain their original structure.

As I mentioned in my earlier report, it is up to authors to write paper rather than reviewers, so I respect the author(s)’ decision here even if I do not, unfortunately, believe it makes their paper particularly citable.

I have no empirical qualms with the evidence presented which is straightforward and transparent.

7. PLOS authors have the option to publish the peer review history of their article (what does this mean?). If published, this will include your full peer review and any attached files.

Reviewer #1: No

Reviewer #2: No

---

## [Author Response · Author response to Decision Letter 1]

14 Mar 2023

Dear PLoS ONE Editor,

First of all, we would like to thank the reviewers for taking the time to review our work and give us their valuable feedback. We have edited the manuscript to address their concerns following the suggestions offered. As a result of this revision, we feel that the outcome has improved again. We answer to each of their comments below and describe the relevant changes that we have made to the manuscript.

Editorial Comment

"1. Please review your reference list to ensure that it is complete and correct. If you have cited papers that have been retracted, please include the rationale for doing so in the manuscript text or remove these references and replace them with relevant current references. Any changes to the reference list should be mentioned in the rebuttal letter that accompanies your revised manuscript. If you need to cite a retracted article, indicate the article’s retracted status in the References list and also include a citation and full reference for the retraction notice."

Answer and Revisions

We have completely revised our Reference List. There are no papers that have been retracted among them nor have we made any additional changes to our Reference List.

Reviewer's Questions

"3. Has the statistical analysis been performed appropriately and rigorously?

Reviewer #1: No

Reviewer #2: Yes"

Response and Revisions

Following R1's suggestions, we have made our description and interpretation of results of Table 2 more precise. As the reviewer indicated, the adjusted R2 is better interpreted in terms of the goodness-of-fit of that model to the DV. Our interpretation of the data can now be read in that way (p.15, lines 318-334). We have also included in Table 2 the AIC (Akaike Information Criterion) and BIC (Bayesian Information Criterion) values for these models (p.15). Since these models are not nested nor do they contain the same number of variables, these measures are appropriate to determine the goodness-of-fit of each and to compare these different models. As can be seen, both values show us a similar ordering of the variables to that of adjusted R2 and allow us to rank them before introducing them into the joint model. 

As for our interpretation of the Negelkerke R2, we have also avoided talking about the percentage of variance explained of the dependent variable (p.17, lines 347-348; p.18, lines 386-387). This information only allows us to assess the fit of the model, and this is how it can be read in the revised version of the paper.

Reviewer's Questions

"5. Is the manuscript presented in an intelligible fashion and written in standard English?

Reviewer #1: No

Reviewer #2: Yes"

Response and Revisions

In order to make the text easier to read and understand, we have clarified the point that had been identified as confusing by R1 (p.17, lines 351-354). In particular, the interpretation of the contribution of the different hypotheses in the discussion has been refined (p.21, lines 441-443; p.21, lines 454-457; p.22, lines 471-473). This new reading is made easier by the changes made to the description and interpretation of Table 2, as mentioned above. In this way, not only is the interpretation of the results obtained more precise, but it also helps the reader to properly follow the explanation given. In addition, some typos have been corrected, which R1 kindly pointed out to us for correction in the revised manuscript.

Reviewer comments

Reviewer #1

"Minor things regarding wording

I find the sentence “all the covariates included in the remaining hypotheses…” (p. 17, line 355) confusing, because there has been no hypothesis discussed briefly before and I struggle to understand which hypotheses, if not all, the authors refer to exactly.

Some of the conclusions are not fully clear to me. In your discussion, you state that “the variables associated with the territorial conflict are those which contribute most to explaining the vote for VOX” (p. 21, line 445-6), but I do not see that reflected in Table 2 nor in the beta coefficients presented in Table A9. Maybe you could rephrase this sentence to reflect that, e.g. anti-feminist attitudes, seem to play an at least equally important role – which I assume was also one of the main arguments you wanted to make. As of know, it reads as a general statement that all else being equal, positions regarding the Spanish territorial conflicts are most important. Similarly, the discussion states that “a rejection of immigration has been the third most important factor” (p. 22, line 475), although this is not reflected in the beta coefficients in Table A9. More than two factors show a larger beta coefficient, e.g. the fight against crime."

Response and Revisions

We agree with R1 that the sentence (s)he points out could be a bit confusing. For this reason, we have integrated the explanation together with the previous paragraph (p.17, lines 351-354). In this way, it does not lose the point of what is being discussed or lead to confusion by referring to the hypotheses. 

Moreover, the interpretation of the specific role of each hypothesis in the discussion has benefited from the fact that we have been more precise with the language when describing and interpreting our results regarding the adjusted R2 in Table 2. In this sense, some sentences that Reviewer 1 pointed out in the discussion section have been modified so that they fully stick to the joint reading of the results of the two tables and its results (Table 2 and Table 3). Both tables show the relevance of the variables relating to the territorial hypothesis (H2.1) (particularly that relating to preferences regarding the territorial structure of the State) and those relating to the feminism hypothesis (H1). Both their fit to the model (Table 2) and their contribution to the explanation of DV (Table 3) indicate their importance. The variables relating to the immigration (H2.2) or authoritarianism hypothesis (H2.3) also show a good contribution to the explanation of the VOX vote. Both tables, moreover, do not support the populism hypothesis (H2.4) as relevant in this explanation. We have therefore been careful to explain the relevance and contribution of all these variables in explaining the VOX vote (p.21, lines 441-443; p.21, lines 454-457; p.22, lines 471-473).

Reviewer #1

"Typos

p. 2, line 37 “these voters [have not been] so [different] from voters of other European ...”

p. 17, line 342-3: “that moderately correlate with otherS indicators”

p. 17, line 345: “does not change [] the results [substantially]”? (English not my mother tongue, so maybe I am wrong here.)

p. 22, line 479: “feminism, [an] issue that VOX …”

p. 19, line 410: “the variable which focus[es]...”

In Table A9, A11 and A12 of the appendix, the authors use the “,” instead of the “.” for their decimals (e.g. 0,877 instead of 0.877)."

Response and Revisions

We have corrected each of these typos pointed out by R1, both in the text (p.2, line 37; p.17, line 343; p.17, line 345; p.19, line 406; p.22, line 466) and in the Supplementary Materials (Table A6, Table A7, Table A9, Table A10, Table A11, Table A12 and Table A13). We would like to thank her/him for providing us with such a detailed review. This feedback has proven invaluable in enhancing the initial version of the paper, and has helped us identify minor details that we had previously overlooked.

Reviewer #1

"Discussion of R² values

I would like to rise one issue that reviewer 2 had previously mentioned as a minor point. Your response and adjustments point out that you want to rank models in Table 2 based on their explanatory power using the adjusted R². From what I learned, the adjusted R², in contrast to the R², cannot be (easily) interpreted as the percentage of variance explained by the model because it does not only take variance into account, but also the number of variables included in the model. While thinking about possible solutions easy to implement during a second round of reviewing, other questions arose:

What are the underlying models for Table 2? Most importantly: which set of variables do they contain? Do they contain control variables or only variables for each hypothesis (feminism, territorial, …)? For each hypothesis, do the models contain the full set of items (Table A5) or only the item included in the regression analyses in Table 3?

Depending on the answers to these questions, the authors might consider 1) using the simple R² and compare the models based on that (if the models contain the same number of variables, and if the set of respondents would be the same in each model), 2) just describing the simple R² as percentage of variance explained but refrain from comparing models, or 3) tone down the language on the percentage of variance explained by the model. If you want to stick to the model comparison, I would recommend to use additional criteria such as AIC and BIC to compare the models with regard to their goodness of fit because the models are not nested nor do they seem to contain the same number of variables.

I can recommend A Guide to Modern Econometrics by Marno Verbeek for these things, which I regularly use myself, especially pages R², AIC, BIC and model comparisons.

I do not think that the model comparison and Table 2 is super important to answer your research question, which is why I would be ok with what-ever solution you find. I would just like to encourage you to be more precise in describing and interpreting the models presented in Table 2 or alternative models you might choose. It would also be great if the authors could add the information regarding the questions above to the supplementary material.

Similarly, I would like to ask the authors to reconsider their interpretation of the Negelkerke R². I have never used it before, but as far as I understood, this measure, again is not based on the variance (only) and thus, should not be interpreted as the percentage of the variance explained by the model. Again, one solution might be to simply delete the statement that this is equal to “explaining 46% of the variability of vote recall” (p. 19, line 389-90) and simply point out that the model fit seems great."

Response and Revisions

We have reviewed in great detail the discussion surrounding the interpretation of R2 values as underlined in the first round of reviews by Reviewer 2 and now by Reviewer 1. Since, as both of them noted, adjusted R2 cannot easily be interpreted in terms of the percentage of explained variance of the DV, in the description and interpretation of Table 2 we have chosen to stick to the explanation that we can give: the goodness-of-fit of each model to the explanation of the dependent variable. Our models are not embedded, nor do they contain the same number of variables (see Table A5 in the Supplementary Materials), so we could not go for the first option suggested by R1. Since our intention was to stick to the model comparison, we opted for the third option (p.15, lines 318-334). As suggested, we have included the values of AIC and BIC in Table 2 (p.15) as additional criteria, in order to strengthen our interpretation and increase the transparency of the comparative analysis of our models.

We have clarified in the text (p.15, lines 321-322) and in the Supplementary Materials (Table A5) which variables compose the different models analysed in Table 2. Thus, the reader can better and more quickly understand the results of each model shown there. 

To make all this clearer, we have also toned down throughout the text the references to the percentage of variance explained by the models. This concerns both the interpretation of the adjusted R2 (p.15, lines 318-334) and the Negelkerke R2 (p.17, lines 347-348; p.18, lines 386-387). The recommended book, "A Guide to Modern Econometrics" by Marno Verbeek, has been really helpful in enabling us to provide a more precise description and interpretation of the adjusted R2 values.

Reviewer #2

"The authors have made some updates to their initial submission, largely in response to some of the concerns raised by reviewer 1.

My initial primary concern with the paper was that the article did not make any original contribution to knowledge based on empirical findings or on theoretical arguments. I made some suggestions about changing the focus of the paper to make it more theoretically rich as opposed to empirically descriptive, but the authors have opted to retain their original structure.

As I mentioned in my earlier report, it is up to authors to write paper rather than reviewers, so I respect the author(s)’ decision here even if I do not, unfortunately, believe it makes their paper particularly citable.

I have no empirical qualms with the evidence presented which is straightforward and transparent."

Response and Revisions

We would like to thank R2 for all the constructive feedback she/he has provided in both review rounds. The discussion regarding R2 has been a pivotal aspect of this round, one which we consider both enriching and gratifying. It has allowed us to expand our personal learning and as a result, the paper has been significantly improved.

Thanks again to all of you for what we feel has been a very productive review process.

---

## [Editor Report · Decision Letter 2]

20 Mar 2023

NOT ONLY A TERRITORIAL MATTER: THE ELECTORAL SURGE OF VOX AND THE ANTI-LIBERTARIAN REACTION

PONE-D-22-22220R2

Dear Dr. Ramis Moyano,

We’re pleased to inform you that your manuscript has been judged scientifically suitable for publication and will be formally accepted for publication once it meets all outstanding technical requirements.

Kind regards,

Jean-François Daoust

Academic Editor

PLOS ONE
---

## [Editor Report · Acceptance letter]

28 Mar 2023

PONE-D-22-22220R2 

NOT ONLY A TERRITORIAL MATTER: THE ELECTORAL SURGE OF VOX AND THE ANTI-LIBERTARIAN REACTION 

Dear Dr. Ramis Moyano:

I'm pleased to inform you that your manuscript has been deemed suitable for publication in PLOS ONE. Congratulations! Your manuscript is now with our production department. 

Kind regards, 

on behalf of

Dr. Jean-François Daoust 

Academic Editor

PLOS ONE